# Structure of the lens MP20 mediated adhesive junction

William J. Nicolas [1,2,5], Anna Shiriaeva [1,5], Michael W. Martynowycz[1,5], Angus C. Grey [3], Yasmeen N. Ruma[1,2], Paul J. Donaldson [3] & Tamir Gonen [1,2,4] ✉

Human lens fiber membrane intrinsic protein MP20 is the second most abundant membrane protein of the human eye lens. Despite decades of effort its structure and function remained elusive. Here, we determined the MicroED structure of full-length human MP20 in lipidic-cubic phase to a resolution of 3.5 Å. MP20 forms tetramers each of which contain 4 transmembrane α-helices that are packed against one another forming a helical bundle. We find that each MP20 tetramer formed adhesive interactions with an opposing tetramer in a head-to-head fashion. Investigation of MP20 localization in human lenses indicate that in young fiber cells MP20 is initially localized to the cytoplasm in differentiating fiber cells but upon fiber cell maturation is inserted into the plasma membrane, correlating with the restriction of the diffusion of extracellular tracers into the lens. Together these results suggest that MP20 forms lens thin junctions in vivo, confirming its role as a structural protein in the human eye lens essential for its optical transparency.

In the lens, terminally differentiated fiber cells are derived from equatorial epithelial cells. These cells undergo a process of differentiation in which they elongate, lose their cellular organelles and nuclei and become internalized by the continual addition of new layers of differentiating fiber cells at the lens periphery[1]. Hence, the lens exhibits a highly ordered spatial gradient of cell age and differentiation, which makes it useful for studying how protein expression patterns change in aging. One such protein is MP20, a lens-specific membrane protein that is a member of the PMP22/EMP/MP20 subfamily of tetraspanins[2–6] and a distant member of the claudin family of junction-forming proteins[7,8]. While MP20 is the second most abundant membrane protein in lens fiber cells[9] and several mutations in MP20 have been shown to result in cataract[10–18], the structure and function of MP20 are unknown. Previous studies have suggested that MP20 may be involved in signaling[19,20], due to its ability to interact with calmodulin, together with the identification of several phosphorylation sites on its cytoplasmic carboxy terminus[20,21]. In peripheral fiber cells, MP20 is localized intracellularly[4,5]. In mature fiber cells, MP20 becomes

entirely membrane-associated[22,23] and in the rat lens this membrane insertion of MP20 correlates with a decrease in the extracellular spaces[22]. Furthermore, MP20 has also been shown to interact with galectin-3, a prominent cell adhesion modulator[24,25]. Together these suggest that MP20 may form adhesive junctions in vivo, although both its structure and function remain elusive.

Past efforts at determining the structure of MP20 yielded low-resolution images that could not identify the MP20 function[26]. The predicted MP20 mass is 22 kDa, but the literature often refers to the protein anywhere between 16–22 kDa based on its appearance on SDS PAGE[24,27]. MP20 is too small for structural characterization using single-particle cryo-electron microscopy (cryo-EM)[27]. For this reason past efforts focused on negative stain electron microscopy and electron crystallography of two-dimensional crystals[26]. These studies showed that MP20 can oligomerize in membranes and form tetrameric assemblies, but its role remained unclear as the attainable resolution was limited, and the crystals were too small and sparse for the available technologies. Efforts using X-ray crystallography likewise failed

[1]Department of Biological Chemistry, David Geffen School of Medicine, University of California, Los Angeles, CA, USA. [2]Howard Hughes Medical Institute, University of California, Los Angeles, CA, USA. [3]Department of Physiology, School of Medical Sciences, University of Auckland, Auckland, New Zealand. [4]Department of Physiology, David Geffen School of Medicine, University of California, Los Angeles, CA, USA. [5]These authors contributed equally: William J. Nicolas, Anna Shiriaeva, Michael W. Martynowycz. ✉e-mail: tgonen@g.ucla.edu

because MP20 crystals were far too small even by today's standards. Such vanishingly small crystals are amenable to structure determination by another more recent cryo-EM method known as Microcrystal Electron Diffraction, or MicroED[28,29]. With MicroED, crystals that are a billionth the size needed for X-ray crystallography can be interrogated by electron diffraction under cryogenic conditions. The crystal is continuously rotated in the electron beam while diffraction is recorded as a movie using a fast camera. This method has recently been enhanced by ion-beam milling with gallium[30,31] or plasma[32] to prepare optimally thick samples- either by making large crystals smaller, or by removing excess material surrounding the microcrystals. This method has successfully elucidated structures such as the A$_{2A}$ adenosine[32,33] and vasopressin 1B[34] receptors by initially tagging proteins with fluorophores, crystallizing them in a lipidic cubic phase (LCP), and using cryogenic fluorescence microscopy for targeting, followed by plasma beam milling and MicroED analysis.

In this work, we express human MP20 heterologously, purify the protein to homogeneity and successfully grow very small crystals of the protein embedded in LCP. By combining fluorescence cryo-plasma Focused Ion Beam (cryo-pFIB) milling and MicroED, we determine the structure of human MP20 at a resolution of 3.5 Å. The structure reveals the overall protein fold, and identifies the role of MP20 in forming cell-to-cell adhesive junctions. Interrogation of the protein in native tissues suggests that in human lenses MP20 forms the 11 nm "thin" junctions, a function that helps support lens architecture and structure.

## Results and discussion

Lens MP20 is a small membrane protein that on SDS PAGE appears as 18 kDa[2]. We expressed human MP20 in SF9 insect cells and purified the protein by immobilized metal affinity chromatography (IMAC) as described in materials and methods. While the protein runs as a single band on reducing SDS PAGE consistent with ~20 kDa, the FPLC trace suggests that it purifies as an oligomer of much larger size (Fig. 1a and Supplementary Fig. 1). Indeed, past reports have indicated that MP20 could form higher oligomers[26]. Considering the micelle size of the detergent that we used for purification, the FPLC trace likely

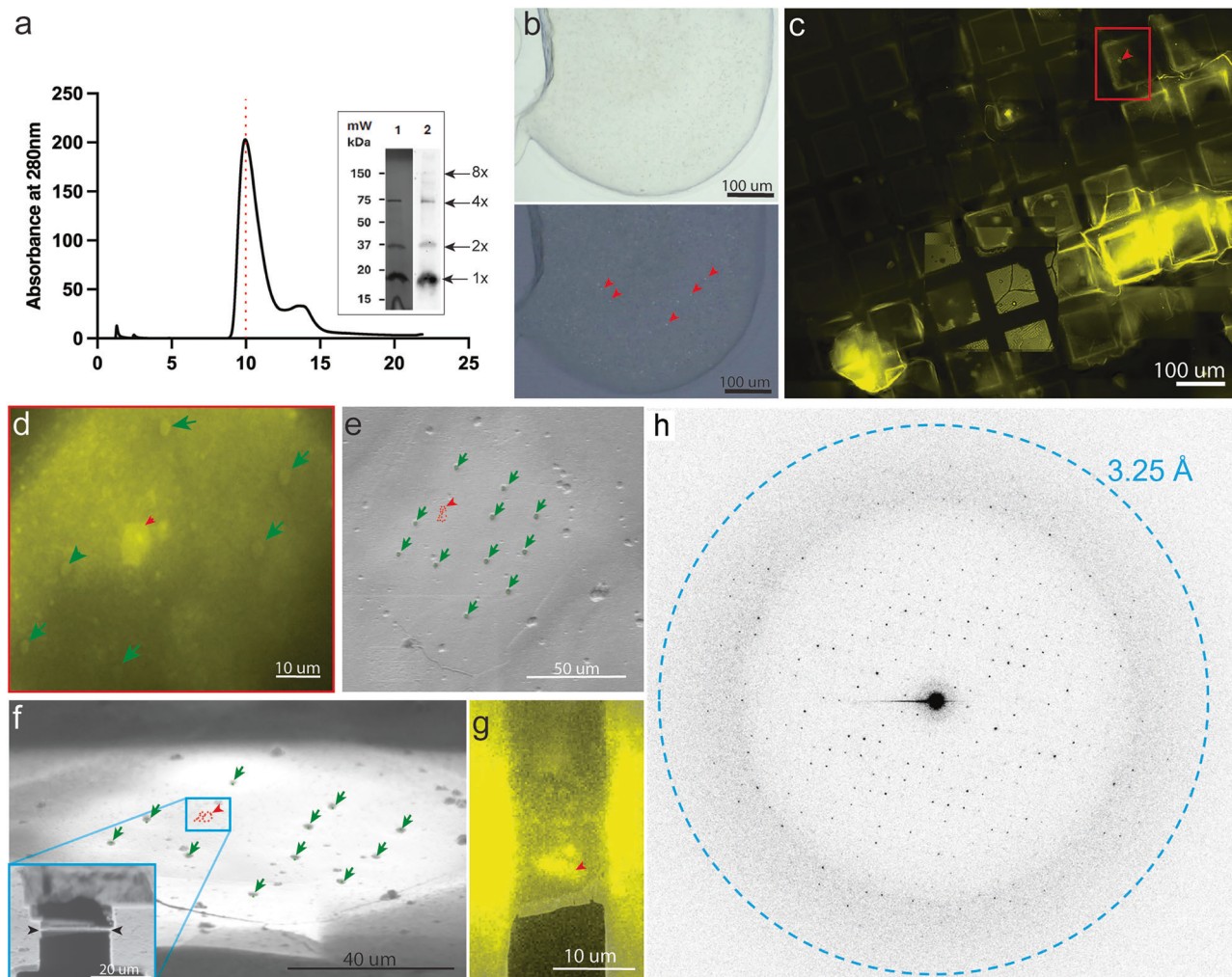

**Fig. 1 | Expression, purification and crystallization of human lens MP20. a** FPLC trace of MP20 with accompanying SDS-PAGE gel (lane 1) and western blot (lane 2). Arrows indicate the MP20 eluted as a uniform peak consistent in size with an MP20 octamer (8 × MP20 plus two detergent micelles). Inset SDS-PAGE of the peak showing purified MP20 and faint bands for its higher molecular weight oligomers (Lane 1) and the corresponding western blot (Lane 2). **b** Cross-polarized light image of a typical drop of LCP with small MP20 crystals appearing as tiny puncta (red arrowheads). **c** Region of a whole-grid fluorescent atlas showing the LCP with JaneliaFluor 550 conjugated crystals (yellow signal). **d** Same crystal boxed out in red in panel c. but imaged in the SEM with the iFLM (red arrowhead). FIBucial markers created with the FIB beam are also visible (green dots and arrows). SEM top-down image (**e**) and FIB grazing incidence image[26] (**f**) of the same field of view (FOV) as in (**d**) with reprojected coordinates of the crystal (red markers and arrowhead) and the FIBucials (green arrowheads). **f** contains shows an inset of the boxed-out region after milling was performed to create a lamella on the targeted crystal. **g** Overlay of the SEM view of the final lamella with the fluorescence signal emanating from the crystal on the milled lamella, confirming presence of the crystal. **h**. An example of a diffraction pattern from the MicroED dataset collected on the crystal lamella shown in (**g**). Blue dashed ring represents diffraction limit for this dataset (3.25 Å). **a**–**g** One representative successful targeted milling example among a series of 25 of them.

corresponds to 8 × MP20 monomers. The purified protein was fluorescently labeled and mixed with lipids for crystallization in LCP[32–35]. We knew that lipids were imperative for this process based on past cryo-EM and electron crystallography studies we conducted on MP20 that was purified from native sources[26]. Some conditions indicated the presence of puncta when screened under cross-polarized light, suggesting that microcrystals of MP20 formed (Fig. 1b, red arrowheads). The sample was deposited onto a cryo-EM grid as described before[32,34] and nanocrystals were initially identified using cryogenic fluorescence microscopy. Grids were thoroughly screened by cryo-fluorescence to identify the best grids and the best grid regions with the most promising crystals (Fig. 1c). The best grids were then transferred into the cryo-pFIB/SEM with an integrated fluorescence microscope inside the chamber (iFLM) that allowed to recall the previously identified areas of

interest (Fig. 1d, red arrowhead). Following targeting and correlative light-EM methods that were described before[32,34] we were able to prepare crystalline lamella of MP20 crystals (Fig. 1e–g) for MicroED analyses.

The grids were then transferred to a cryo-TEM for MicroED analyses. Using a highly attenuated electron source, an energy filter, and a sensitive direct electron detector we were able to collect 3.5 Å resolution MicroED data (Fig. 1h, Supplementary Movie 1). While the reflections appeared strong at the beginning of the data collection, the intensity quickly dropped, likely due to the accumulation of radiation damage[36]. Indexing the data indicated that the crystal symmetry was $P$ 4 2 2 with unit cell dimensions of $a = b = 56.18$ Å and $c = 142.50$ Å. Typically a single nanocrystal would provide sufficient information to cover approximately 60° of reciprocal space so several datasets originating from 6 different nanocrystals had to be merged to increase completeness to 86.6% (Table 1). Phasing of the MP20 data was accomplished using a poly-alanine model of an MP20 monomer generated by ColabFold[37], a cloud-based implementation of alphafold2[38], since no other experimental MP20 homologous structure was available at the protein data bank. Phasing results indicated a true space group of $P$ 4 $2_1$ 2 with a single copy in the asymmetric unit. This creates an arrangement of four monomers around a central axis with an additional four monomers coordinating above it, creating an overall octamer (Supplementary Fig. 2).

MP20 has the overall topology consisting of four transmembrane helices (Ser3-Ala22, Ala59-Phe86, Ser100-Phe126, and Trp138-Cys166) connected by two extracellular loops, ECL1 and ECL2 (Met23-Ile58 and Leu127-Ser137), and one intracellular loop (Ala87-Phe99), with both its N and C termini localized intracellularly (Fig. 2a, Supplementary Movie 2). This topology appears to be common in tetraspanin and the broader claudin family[39]. A long ECL1 domain connecting helices 1 and 2 is folded in the extracellular side of the membrane in an anti-parallel fashion presenting two subdomains that we name ECL1a and ECL1b. The four alpha-helices form a tight bundle, in contradiction with past reports suggesting that MP20 might form a channel[2,40]. A space-filling model of MP20 indicated that no cavities or channel/path exists in a single MP20 monomer (Fig. 2b, c). While it is possible that, at this resolution, we are not resolving the channel path, we note that for another lens membrane protein, namely aquaporin-0 (AQP0), even at a similar resolution a clear channel was observed[41]. These analyses and

### Table 1 | The MicroED structure of human MP20

| | |
|---|---|
| Wavelength (Å) | 0.0197 |
| Resolution range (Å) | 52.26–3.5 (52.26–3.5) |
| Space group | $P$ 4 $2_1$ 2 |
| Unit Cell ($a$, $b$, $c$) (Å) ($\alpha$, $\beta$, $\gamma$) (°) | 56.18, 56.18, 142.5 90, 90, 90 |
| Total reflections (#) | 58259 (1931) |
| Multiplicity | 20.7 (11.3) |
| Completeness (%) | 86.6 (78.4) |
| $<I/\sigma(I)>$ | 7.19 (2.32) |
| R-meas | 0.39 (1.57) |
| R-pim | 0.087 (0.370) |
| $CC_{1/2}$ (%) | 98.9 (38.0) |
| R-work (%) | 0.3316 |
| R-free (%) | 0.3510 |
| Number of atoms (#) | 1323 |
| RMS bonds | 0.002 |
| RMS angles | 0.34 |
| Ramachandran (F/A/O) (%) | 83.54/14.02/2.44 |
| Rotamer outliers (%) | 0.00 |
| Clashscore | 5.34 |
| Average B-factor (Å$^2$) | 88.55 |

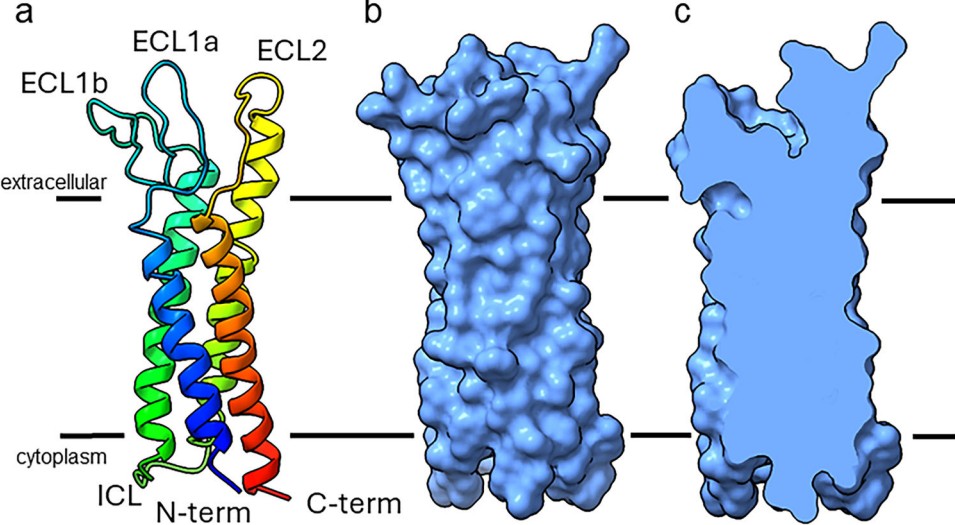

**Fig. 2 | MicroED structure of the lens MP20. a** Structure of lens MP20 in rainbow with the N terminus in blue and C-terminus in red. Both termini are cytoplasmic. The loops are indicated as ECL1a, ECL1b, ECL2 and ICL. **b, c** Space-filling model (half is clipped in **c**) of the MP20 monomer showing that no channel or pathway can be seen in the protein. The black lines indicate the position of the lipid bilayer with the extracellular and intracellular sides as indicated.

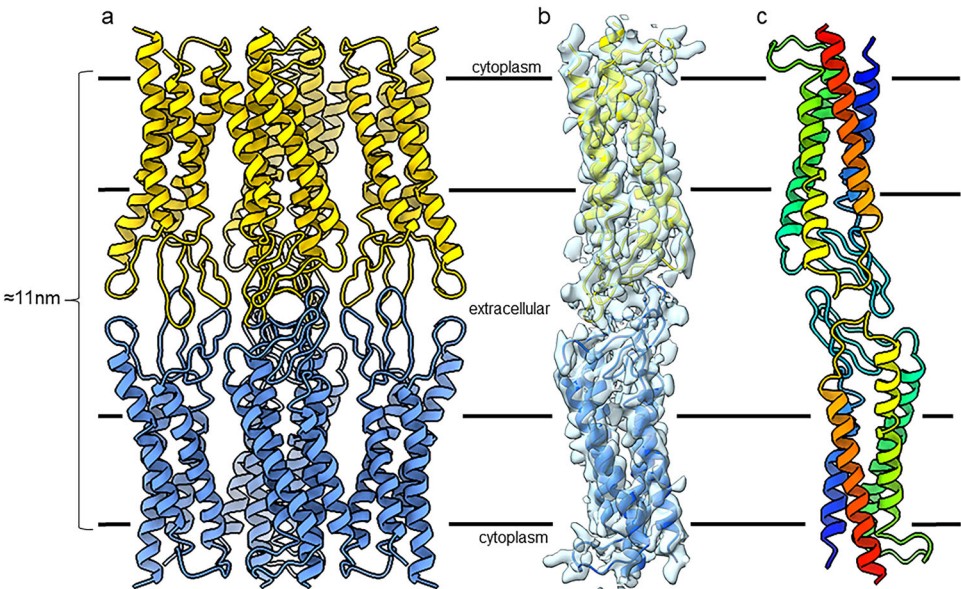

**Fig. 3 | MicroED structure of the lens MP20 junctions. a** MP20 mediated membrane junction. Two MP20 tetramers interact in a head-to-head fashion (yellow and blue). The adhesive interactions are mediated by the extracellular loops. **b** MicroED density map around one yellow and one blue MP20 monomer as they form the adhesive interaction. **c** Two monomers shown in rainbow with the N terminus in blue and C terminus in red. The black lines indicate the position of the lipid bilayer with the extracellular and intracellular sides as indicated.

observations suggest to us that an MP20 monomer in this case does not form a channel, although it might form a proton channel which would not require a soluble path but rather a proton wire[42]. However, we could not locate a clear path for a continuous proton wire to form from the cytoplasmic to the extracellular side of MP20. Past electron crystallography analyses of reconstituted 2D crystals of MP20 indicated that the protein might assemble into tetramers[26]. Indeed, we can find an MP20 tetramer in our crystals. A close look of MP20 tetramers likewise does not identify any selectivity or path for substrates. This is again consistent with MP20 not forming a channel in this conformation.

When observing the full unit cell in our crystals it became clear that MP20 forms adhesive junctions. One MP20 tetramer was found interacting with an opposing MP20 tetramer from the opposing bilayer (Supplementary Fig. 2). The interaction is mainly mediated through the loop regions of each MP20 monomer (Supplementary Fig. 3). ECL1a, ECL1b, and ECL2 form an adhesive interaction resembling a handshake with the ECL1a and ECL1b and ECL2 from the opposing monomer. The interactions appear to be mainly electrostatic packing as several charged and hydroxylated residues can be seen forming adhesive interactions (Fig. 3b, c, Supplementary Fig. 3, Supplementary Movies 3–6). The ECL1b has cysteine residues Cys46 and Cys51 that are in close proximity and can potentially form a disulfide bond in vivo. These cysteine residues are conserved within the PMP22/EMP/MP20 family indicating that they may be functionally important (Supplementary Fig. 3, and Supplementary Movie 3, yellow dashed pseudobonds). In ECL1a residue Tyr52 is located at the interface, and in close proximity with Asn49 of the ECL1b of another MP20 molecule in the opposing bilayer. Lys50 in the same loop is directed towards a neighboring molecule in the same lipidic bilayer (Supplementary Fig. 3C, green dashed pseudobonds). ECL1b has polar residues Ser34, Ser36 that are directed towards the interface with the Thr55 in ECL2 of the opposing molecule. The smaller ECL2 loop, connecting transmembrane helices 3 and 4, has two consecutive arginine residues Arg129 and Arg130 involved in the contact with the ECL2 loop from another MP20 molecules. Phe131 in ECL2 is forming hydrophobic interactions with Tyr31 in ECL1b, Leu127 in ECL2, and Leu33 in the opposing molecule. The total thickness of the MP20 mediated

membrane junction is approximately 11 nm (Fig. 3a), which is consistent with the lens "thin" 11 nm junctions that have been described for decades[43].

Next, to assess whether MP20 forms junctions in vivo, we utilized an optimized fixation and sectioning protocol that allows sections through all regions of the human lens to be labeled with MP20 antibodies and imaged using fluorescence[44]. Past studies using this approach have identified MP20 throughout the lens and showed that its subcellular location changed during lens development. Using wheat germ agglutinin (WGA), we labeled cell membranes in a human lens (Fig. 4a). Three areas can be easily identified: a single layer of epithelium, which is followed by the fiber cells that make the bulk of the lens. The fiber cells in the outer cortex are arranged neatly in hexagonal arrays while deeper into the lens toward the core the fiber cells undergo remodeling and maturation, and the hexagonal packing is lost. The subcellular localization of MP20 changes during these maturation events (Fig. 4b). Initially, MP20 is found intracellularly as puncta inside the fiber cell cytoplasm, not on the plasma membrane. This is consistent with past studies that were done on lenses from rodents[22,25]. Deeper in the lens, MP20 labeling shifts from the intracellular cytoplasmic puncta to the plasma membrane. In the adult nucleus, all MP20 appears to be inserted into the plasma membrane.

We next asked whether adhesive junctions form upon MP20 insertion to the plasma membrane. We incubated the same lenses with Texas red Dextran which is known to penetrate lenses via the extracellular spaces[44]. Indeed, this analysis indicated that labeling was observed between lens fiber cells in the outer cortex of the human lenses (Fig. 4c). The dye still penetrated deeper into the lenses until it abruptly stopped right after the remodeling zone (Fig. 4c, white line). An overlay of the MP20 labeling with the extracellular marker is presented in Fig. 4d. This overlay indicates that upon insertion of MP20 into the plasma membrane of the lens fiber cells, the extracellular spaces between cells is constricted to the point where Texas red dextran could no longer diffuse. This restriction in the extracellular space does not appear to be diffusion limited, since increasing the incubation time did not increase the depth of Texas Red-dextran penetration past the adult nucleus region (Fig. 4c, dotted line)[44]. This indicated that dye diffusion into the lens through the extracellular

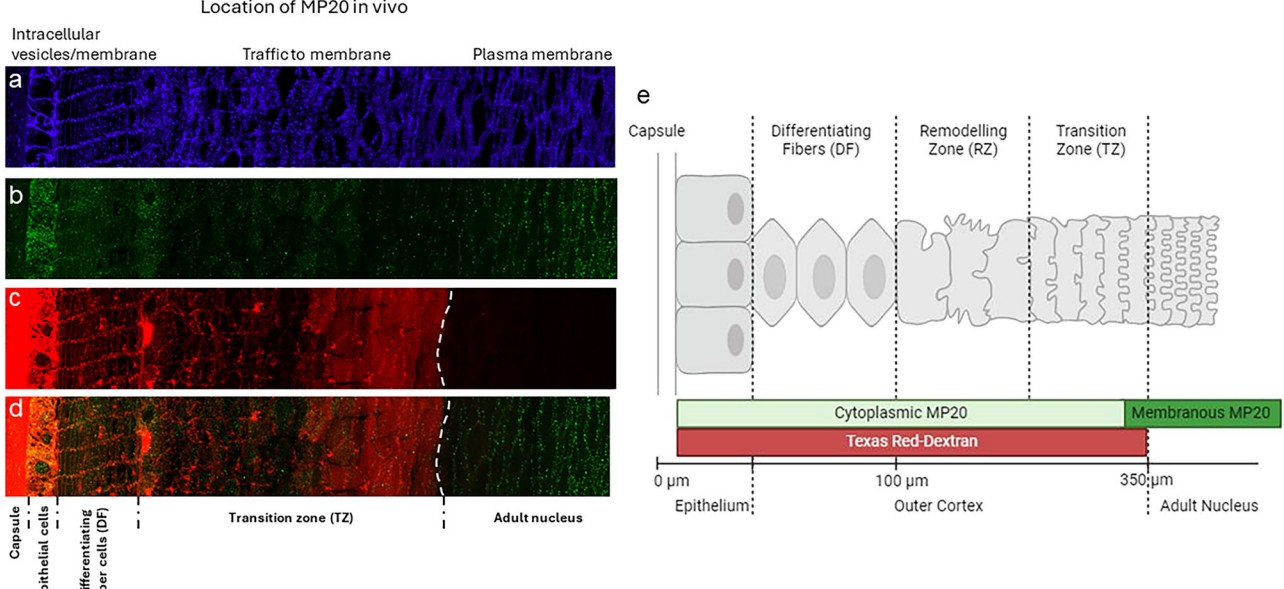

**Fig. 4 | Insertion of MP20 into fiber cell membranes correlates with the constriction of the extracellular space in vivo.** Human lens labeled with **a** WGA **b** MP20 **c** extracellular dye. **d** Overlay of (**b**, **c**). Image montage of the outer cortical region of cryosection taken through the equator of a human donor lenses organ cultured in the extracellular space marker Texas Red-dextran for 6 h. **a** WGA-TRITC labeling to highlight the change in membrane morphology as fiber cells differentiate and become internalized into the adult nucleus. **b** MP20 labeling showing the shift from the cytoplasm to the membrane. **c** Texas Red-dextran labeling of the extracellular space. White line showing where the dye stops and the extracellular spaces become restricted. **d** Double labeling of MP20 (green) and Texas Red-dextran (red) showing the formation of the extracellular diffusion barrier corelates with the membrane insertion of MP20 into the membranes of fiber cells in the adult nucleus. **e** Schematic diagram (not drawn to scale) summarizing changes in the morphology of differentiating fiber cells in the outer cortex of the human lens. DF differentiating fiber cells, RZ remodeling zone, TZ transition zone.

space was not limited by diffusion velocities but by a physical barrier that coincided with the insertion of MP20 into the membrane (Fig. 4d). Together these results indicate that MP20 forms adhesive junctions between cells in vivo and that these junctions restrict the extracellular spaces in a fashion reminiscent of tight junctions.

As the MicroED method matures and more technologies are developed and optimized for dealing with difficult samples, novel structures that were beyond the reach of other structural biology methods will be determined experimentally. In this case, lens MP20 was subjected to intensive structural biology efforts for several decades by several laboratories worldwide employing all known structural biology methods including freeze fracture, thin section electron microscopy, single particle cryo-EM, electron crystallography, X-ray crystallography and even attempts using NMR, and AFM. In all these cases, only anecdotes were obtained and the MP20 enigma remained unsolved for half a century.

The MicroED structure presented here is the first experimentally determined structure of MP20, and more largely of any claudin-like protein involved in a tight junction, proving that in a lipid membrane MP20 can form cell-to-cell adhesive junctions. While each MP20 traversed the membrane 4 times, the protein assembled into tetramers in membranes and its extracellular loops formed adhesive interactions with an apposing MP20 in a head-to-head fashion. The interaction is best described as a handshake. The length of this junction is consistent with Lens thin 11 nm junctions, which have been known to also contain AQP0 junctions[41,43]. However, in the case of MP20 it is still unknown whether the protein can also form a channel under some circumstances.

MP20 is a small membrane protein that belongs to the PMP22/EMP/MP20 family and is the first to be structurally characterized. This family is distantly related to the claudin superfamily of proteins that are known to form tight junctions[45]. While all these proteins contain 4 transmembrane helices, MP20 is the first to be determined in membranes and indeed the first to pack in the crystals as a junction. Small

membrane proteins like MP20 are beyond the reach of single particle cryo-EM due to their size which is 3× smaller than the physical limit for cryo-EM (approximately 50 kDa)[46]. Recent work using single particle cryo-EM on claudin-4 inserted into micelles and bound to *Clostridium perfringens* enterotoxin were complexed with sFab to increase their size and facilitate imaging[47,48]. Even with this approach a relatively low-resolution image of the claudin was obtained and no junction observed since the Fab prevented junction formation. This illustrates the importance of solving membrane protein structures in the context of the lipid membrane rather than in detergents, and without the use of Fabs.

We also showed that the insertion of MP20 into cell membranes coincides with a constriction of the extracellular space in vivo in human lenses. It is presumed that MP20 therefore contributes to the structural stability of lens tissue and maintenance of transparency throughout lifetime. Future studies should delve into the structure of MP20 in its tetrameric form and analyze whether the protein can associate with other proteins and lipids to form channel complexes as has been observed in our previous lower resolution studies[26]. Further, a variety of congenital cataracts are associated with MP20 mutation[15], but it is not clear what the structural impact of these mutations may have on the protein and how they may affect its function.

While predictive methods like Alphafold[49] are increasing in popularity, the study presented here serves an important reminder to always validate models with experiments and, if possible, to experimentally determine structures. While Alphafold provided us with a reliable molecular replacement model of an MP20 monomer for phasing, even Alphafold3[49] which was marketed as a server for predicting protein interactions failed in predicting the oligomeric assembly of lens MP20 (Supplementary Fig. 4). When asked for predictive models for a dimer or for the octamer nonsensical assemblies were predicted, though with a low degree of certainty. In this case, since MP20 monomer is similar to the previously determined claudin[47] and an experimental structure exists in the protein data bank, the

predicted model for the monomer was essentially correct. However, since the adhesive junction was never observed experimentally it is not surprising that the predictive power of was limited. This serves as a reminder of the value of an experimental approach to structure determination and the limitation predictive methods still have when dealing with novel assemblies. Methods like MicroED will play a pivotal role in unraveling the biology behind MP20 and other small integral membrane proteins that are difficult to study with the more widely spread X-ray crystallography and cryo-EM techniques.

The implications of this study extend beyond the structural elucidation of MP20. By Identifying MP20 as a component of the thin lens junctions, this work provides insights into the architecture of the lens and its impact on lens transparency. The close apposition of fiber cells, facilitated by these junctions, minimizes light scattering and maintains the lens's optical properties. This is in line with theoretical calculations stating that transparency can be maintained as long as the extracellular junctions are less than 30 nm, as well as experimental evidence linking increased extracellular junction thickness with the emergence of cataract[50]. MP20's role in junction formation has significant implications for cataract research. Given the association between MP20 mutations and congenital cataracts, understanding the structural basis of these junctions opens avenues for investigating how these mutations disrupt lens development and transparency. Future studies involving mutagenesis of key interface residues identified in this structure could pinpoint the molecular mechanisms underlying cataracts. Moreover, electrophysiological studies could probe the possibility of MP20's involvement in channel activity, as suggested by previous lower-resolution studies[26]. This research also contributes to our broader understanding of tight junctions, as MP20 is the first member of the PMP22/EMP/MP20/claudin-like family to be structurally characterized in a membrane environment, forming intercellular junctions within the crystal lattice. This highlights the importance of studying membrane proteins within a lipidic environment to understand their true biological function.

It is important to note that while AlphaFold predictions provided a reasonable model for the MP20 monomer, even the latest iteration of AlphaFold3[49], designed to predict protein complexes, failed to accurately model the tetrameric assembly and inter-tetramer interactions observed in our MicroED structure. This underscores the critical role of experimental structure determination in revealing novel biological assemblies and validating computational predictions.

## Methods

### Construct design
Several constructs with various positions of tags were screened. The construct selected for crystallization due to successful expression and sufficient yield contained the following N-terminal tags: Hemagglutinin precursor peptide fragment MKTIIALSYIFCLV, FLAG (FADYKDDDDAK), LQTM, $His_{10}$, PreScission cleavage site (LEVLFQ) (Supplementary Fig. 1). The construct used for the Fast Protein Liquid Chromatography (FPLC) is as follows: full-length MP20 cloned to the plasmid pcDNA3.1(+) (Genscript) with a C-terminal his tag and flag tag.

### Protein expression and purification
MP20 constructs were expressed in sf9 insect cells using the Bac-to-bac baculovirus expression system (Invitrogen). Cells with a density of $(2–3) \times 10^6$ cells mL$^{-1}$ were infected with baculovirus at 27 °C at a multiplicity of infection of 10, harvested by centrifugation 48 h post-infection and stored at −80 °C until use. Cells were lysed with hypotonic buffer (10 mM HEPES pH 7.5, 10 mM $MgCl_2$, 20 mM KCl). The membrane fraction was isolated from 2000 mL of biomass using Dounce homogenization and ultracentrifugation (25 min, 43000 g, 4 °C). The membranes were subsequently washed with high-salt galactose-containing buffer (1 M NaCl, 10 mM HEPES pH 7.5, 10 mM $MgCl_2$, 20 mM KCl, 100 mM galactose) and isolated by

ultracentrifugation (30 min, 43,000 × $g$, 4 °C). Washed membranes were incubated in the hypotonic buffer in the presence of 2 mg mL$^{-1}$ iodoacetamide for 20 min. MP20 was solubilized from membranes in a volume of 200 mL (4 × 50 mL) by addition of 2× solubilization buffer (100 mM HEPES pH 7.5, 250 mM NaCl, 1% (w/v) n-dodecyl-β-D-maltopyranoside (DDM, Anatrace) and 0.1% (w/v) cholesterol hemisuccinate (CHS, Sigma-Aldrich), 0.1% n-nonyl-β-D-glucopyranoside (NG, Anatrace)) for 2.5 h at 4 °C. Unsolubilized membranes were separated by centrifugation (60,000 × $g$, 50 min). The supernatant was incubated overnight with 1 mL of Talon (immobilized metal affinity chromatography, IMAC) resin (Takara) in the presence of 10 mM imidazole. The sample was washed on a gravity column (Bio-Rad) with 10 column volumes (cv) of wash buffer #1 (50 mM HEPES pH 7.5, 800 mM NaCl, 25 mM imidazole pH 7.5, 10% (v/v) glycerol, 10 mM $MgCl_2$, 0.05%/0.01% (w/v) DDM/CHS) followed by 5 cv of wash buffer 2 (50 mM HEPES pH 7.5, 800 mM NaCl, 10% (v/v) glycerol, 50 mM imidazole, 0.025%/0.005% (w/v) DDM/CHS). Fluorescent labeling was carried out on column using Janelia Fluor 549 NHS-ester dissolved in DMF (4 mg mL$^{-1}$) and added in concentration 0.1% (v/v) to the labeling buffer (50 mM Hepes pH 7.5, 800 mM NaCl, 10% (v/v) glycerol, 0.025%/0.005% (w/v) DDM/CHS). Talon with MP20 was incubated with 2 cv of dye-containing labeling buffer for 3 h, then washed cv by cv with 10 cv of the labeling buffer without dye and 3 cv of wash buffer #2. The sample was eluted in 500 µL fractions using elution buffer (50 mM HEPES pH 7.5, 800 mM NaCl, 250 mM imidazole pH 7.5, 10% glycerol, 0.025%/0.005% (w/v) DDM/CHS). The sample was analyzed by HPLC (Shimadzu) (Supplementary Fig. 1) using a size exclusion column (Nanofilm SEC-250 Sepax). Fractions protein in micelles were concentrated to 20 µL using 100 kDa molecular weight cut-off protein concentrators (Amicon Ultra) at 1000 × $g$ for 4–6 h.

For the FPLC experiments, the MP20 construct was expressed in Expi293 cells (Thermo Fisher scientific) using the method described by Pleiner et al.[51]. The cells were grown in Expi293 expression media (Thermo Fisher scientific) to a density of 2 to 3 million cells /ml and then transfected with the MP-20 containing plasmid purified using a plasmid purification Giga kit (Qiagen). PEI MAX 40k (Polysciences, USA) was used as the transfection reagent. Cells were grown at 37 °C, 8% $CO_2$ with 125 rpm and harvested after 68 to 72 h followed by lysis in a sonicator with a lysis buffer (20 mM Tris, pH 7.5, 300 mM NaCl, 10 mM $MgSO_4$, 10% glycerol, DNAse, protease inhibitor)[52]. Cell membranes were then separated by ultracentrifugation (12,000 × $g$, 1 h, 4 °C). MP20 was extracted from the cell membranes by incubating in solubilizing buffer (20 mM Tris, pH 7.5, 300 mM NaCl, 10 mM $MgSO_4$, 10% glycerol, 1% DDM) at 4 °C for 1 h. The supernatant with the protein was then separated by ultracentrifugation (100,000 × $g$ rpm, 45 min, 4 °C) and incubated overnight with Talon metal affinity resin (Takara Bio). The protein was affinity purified from the solubilized fraction in a gravity column (BioRad) using 300 mM imidazole in elution buffer (20 mM Tris, pH 7.5, 150 mM NaCl, 10% glycerol, 0.05% DDM) and further purified by SEC using a Superdex 200 10/300 GL column (GE Healthcare Life Sciences). Protein fractions were then analyzed on a SDS-PAGE gel (BioRad) and detected by western blotting using HRP-conjugated anti-his antibody (Genscript).

### Crystallization of MP20 in LCP
Lipid cubic phase (LCP) was prepared by mixing concentrated protein in the ratio 2:3 (v/v) with molten lipid mix (90 % w/w monoolein and 10 % w/w cholesterol) using a syringe coupling system (Hamilton). Protein concentration was 15 mg mL$^{-1}$. Crystallization was carried out using NT8 robotic dispensing system (Formulatrix). 40 nL LCP drops were dispensed at 85 % humidity and covered with 400 nL of precipitant solution. The plate was sealed and stored at room temperature. Crystals were detected in conditions with 25–100 mM ammonium sulfate, 100 mM ADA pH 5.6–6.0, 27–34 % PEG-400. Crystals appeared after 12 h. Crystals were harvested using 20–50 µm nylon loops. Regions of

the LCP drops were scooped and gently smeared across pre-clipped Cu200 R2:2 grids (Quantifoil) under a humidifier. Grids were flash-frozen in liquid nitrogen and stored frozen until use.

## Whole-grid fluorescence mapping

Shortly after vitrification, grids were batch-screened by cryo-Fluorescence Light Microscopy (cryo-FLM) using the Leica Thunder (Leica Microsystems, model DM6-FS) and the LAS X software (Leica Microsystems, version 3.7.6). For MP20 crystals labeled with Janelia Fluor 549, whole-grid 3D atlases were acquired for all frozen grids in fluorescence (Y3 filter cube−533−558 nm excitation/565 nm dichroic/570−640 nm emission). Several grids were also screened using reflection and/or transmitted light mode to allow for better correlation. 3D atlases were then carefully screened in LAS X after computational cleaning and deconvolution. Regions of interest were labeled. Labels and atlases were then exported for later use in MAPS software (ThermoFisher).

## Plasma FIB milling of deeply embedded MP20 microcrystals

The grids selected for milling were inserted in the ThermoFisher Hydra FIB-SEM. The MAPs software (version 3.22) was used to make SEM atlases of the grids. The previously exported LAS X fluorescence atlases were imported into the MAPs software, including the labels and a fluorescence *in-focus* image. Both were realigned to the SEM atlas, allowing to target in *X-Y* the different crystals labeled during the prior step. The iFLM was then used to confirm presence of crystal and correlate its position as described below.

## In-chamber fluorescent monitoring of crystals

The in-chamber Fluorescence Light Microscope (iFLM), from the original design of the PIEscope[53] was used to do the X-Y-Z correlation of the crystals to be milled (see below). It is situated coaxially to the FIB gun. Our iFLM system has an Olympus UPLZAPO 20× objective (0.8 numerical aperture, and working distance of 0.6 mm), a Basler acA3088-57 detector with a physical pixel of 2.4 μm and an unbinned pixel size of 120 nm. The light source is a 4-LED system from Thor Labs (385, 470,565 and 625 nm) working in conjunction with an epi-illuminator (Thor labs−WFA2001). The filter cube used is a multi-band filter-set optimized for DAPI, FITC, TRITC and Cy5 (Semrock LED-DA/FI/TR/Cy5-B-000). It was operated with the Fluorescent Microscope Control v0.7 software (ThermoFisher). Image acquisition settings were the following: Bin 2 (240 nm pixel size), 565 nm excitation wavelength at 100%, 100 ms exposure time. When Z-stacks were acquired, the Z-step was set to either 0.5 or 1 μm for optimal compromise between fine Z-slicing and time to acquire stack. 3D fluorescent volumes were saved as.tiff stacks.

## FIBucial correlation method

Normal-incidence FIB milling was used to create milled holes surrounding the sample (FIBucials). Beam settings were Ar−5 s unblanking time−7.6 nA−30 kV. The area was then screened by iFLM and fluorescence of FIBucials was checked. When enough clearly fluorescent FIBucials were created (-5), a fluorescent stack with Z-step 1 μm was acquired. A normal-incidence and grazing incidence FIB (milling angle) image of the same area were then acquired where the FIBucials are also visible. The stack was then pre-processed using a custom Jupyter notebook. This entailed reslicing of the stack to have cubic pixels, normalization of intensities with the 3D correlation tool box (3DCT – https://github.com/hermankhfung/3DCT and https://github.com/hermankhfung/tools3dct)[54] library. Then, deconvolution was performed with the Red Lion Fish deconvolution (https://github.com/rosalindfranklininstitute/RedLionfish) library. This allowed to increase the signal-to-noise ratio, especially important for correct Z-estimation of the FIBucials and the crystal[32]. Next custom Jupyter Notebook took the pre-processed stack and opened it with Napari software (https://github.

com/napari/napari). The *x*, *y*, *z* coordinates of the FIBucials are registered by adding a New Points layer. The Z coordinate is where the FIBucial is seen the sharpest. The coordinates of the crystal are registered the same way and the layer of points is saved. The x and y coordinates of the FIBucials on the FIB/SEM images are registered and saved the same way. This script then converts the Napari-generated coordinates to fit the 3DCT coordinate system and the 3DCT GUI is started with the fluorescent stack and the FIB/SEM image along with their list of coordinates. The software is then used to predict the position of the crystal on the FIB image and compute the error of the prediction. An error of <10 pixels was sufficient to prevent over milling of crystal.

## FIB milling of crystals

All milling was done with the Argon beam at 30 kV. In between each milling step, the iFLM was used to assess the presence and integrity of the targeted crystal. The predicted position of the crystal done with the method described above allowed us to make an informed decision as to where to position our initial trenches. Initial trenches were spaced 15 μm apart with the targeted crystal centered in the initial slab. This step was performed at 7.6 nA, a box depth ranging from 20−40 μm and a box width of 20 μm. Box height depended on the topology of the region. What is important is that the trenches clear out everything above and below the initial slab, meaning un-obstructed holes visible in FIB and SEM. Next step consisted in rough milling down to a 5 μm thick slab at 2 nA. Because the Z-resolution provided by our iFLM ranges between 5 and 7 μm, we considered that our predicted crystal position was no-good below 5 μm. We therefore proceeded by shaving only the top of the lamella with 10 μm wide and 0.5 to 1 μm high milling boxes and checking the presence of the crystal with the SEM beam in immersion mode, using the through lens detector at 1.2 kV. When the crystal became visible by contrast difference, the bottom of the lamella was milled until final 200−300 nm thickness. 0.74 nA, 0.2 nA and 60 pA currents were used until lamella was 1 μm, 500 nm and 200−300 nm thick, respectively. The top of the lamella was then polished very lightly at 20 pA for -30 s, presence of fluorescence on the lamella is checked one last time before immediate transfer to Krios for MicroED data acquisition.

## MicroED data collection

Milled grids were loaded into a cryogenically cooled Titan Krios G3i transmission electron microscope operating at an accelerating voltage of 300 kV. All grid atlases were taken at 155 × magnification using SerialEM[55] to identify milled lamellae on the grid. Each lamella was brought to eucentric height. The crystals were not visible using low or high SA imaging at low dose due to the low contrast between the crystal and the surrounding LCP. The crystals were located by initially correlating the iFLM fluorescence and the SEM low-voltage images with the TEM low SA (3600×) images. Optionally, a custom SerialEM script was used to screen sub-areas of the lamellae in 2 μm steps in a grid pattern to find the best diffraction. Continuous rotation MicroED data were collected using an in-house dedicated SerialEM script, on a Falcon 4i direct electron detector in electron counting mode using the electron event representation format (.eer)[56]. A single dataset was collected from each crystal over a wedge of 60 degrees over 420 s with a nominal camera length of 2500 mm. The total exposure for each dataset was approximately 1 e⁻ Å⁻². In order to keep the fluence low enough for counting mode, the beam was spread to a diameter of 30 μm, and then a selected area aperture with a 1 μm diameter was used to select only the diffraction from the crystal was used during the data collection.

## Structure determination and refinement of the MP20 structure

Individual movies in EER format were converted to SMV format using MicroED-tools mrc2smv (available from cryoEM.ucla.edu)[57]. EER files contained approximately 130,000 individual event frames after taking dark frames into account. SMV frames were created by summing 308

individual EER frames, or 1 s of counting data, correcting for post-counting gain, and binning the data by 2 in both the $x$ and $y$ dimension. In this way, each 420 s EER movie resulted in 420 2048 × 2048 SMV formatted images containing 16-bit unsigned integers. MicroED data on SMV frames were processed in XDS[58] where the data were indexed and integrated. Datasets were scaled using XSCALE. The data were indexed in space group P 4 2 2 (#89) with unit cell $(a, b, c)$ (Å) = (56.18, 56.18, 142.50) and $(\alpha, \beta, \gamma)$ (°) = (90, 90, 90). The resolution was cut to 3.5 Å, where the $CC_{1/2}$ value fell to a value of approximately 30%. The structure of MP20 was determined by molecular replacement. The search model was an Alphafold[38] structure truncated to a polyalanine chain. The molecular replacement search was accomplished using PHASER[59] searching in all possible space groups, where the final model was placed in space group P4 $2_1$ 2 (#90) with a Translation Function Z-score (TFZ) of 20 and Log Likelihood Gain (LLG) of 700. The model was initially refined in PHENIX.REFINE[60] using electron scattering factors with group B-factors and rigid body refinement. The sidechains were docked into the trace using the known sequence. The updated model was refined iteratively using COOT[61] and PHENIX.REFINE until the model could no longer be improved. The final R-work and R-free were 33.16% and 35.10%, respectively.

### Human lenses

Lenses of a range of ages were obtained from donor eyes from the New Zealand National Eye Bank within 24-48 h of death. All human lens work was conducted in compliance with the Declaration of Helsinki and was approved by the Northern X Regional ethics Committee (Ref: NTX/07/08/079). 3 human lens was used for the fluorescent study of the insertion of MP20 in the membrane of lens cells (Fig. 4). Biological replicates = 3 human lenses (63 y M, 46 y M, 71 y M) were used for the labeling. Corneas were removed for transplantation and the lens removed and immediately assessed by dark and bright field microscopy and then either fixed immediately for immunohistochemical experiments or first incubated in Artificial Aqueous Humor (125 mM NaCl, 0.5 mM $MgCl_2$, 4.5 mM KCl, 10 mM $NaHCO_3$, 2 mM $CaCl_2$, 5 mM glucose, 20 mM sucrose, 10 mM HEPES, pH 7.2–7.4, 300 ± 5 mOsmol) that contained 1.25 mg/ml of Texas red-dextran (Molecular Probes, Eugene, OR, USA), for 6 h or 24 h before being fixed as described below.

### Immunohistochemistry

Whole lenses were fixed for 24 h in 0.75% paraformaldehyde in PBS, encased in 6% agarose and then either cut in half perpendicular (equatorial) or parallel (axial) to the optic axis using a sharp blade. The halved lenses were fixed for 24 h in 0.75% paraformaldehyde and then cryoprotected by incubation in 10% sucrose-PBS for 1 h, 20% sucrose-PBS for 1 h and then 30% sucrose-PBS overnight at 4°C. For sectioning, half lenses were mounted in an equatorial or axial orientation on pre-chilled chucks and encased in Tissue-Tek O.C.T compound. Lenses were cryosectioned into 12-14 μm thick sections and transferred onto poly-lysine coated microscope slides (Superfrost Plus; ESCO, Electron Microscopy Sciences, Fort Washington, Pennsylvania. USA). Sections were washed three times in PBS and then incubated in blocking solution (3% BSA, 3% normal goat serum (NGS)) for 1 h to reduce non-specific labeling. Sections were labeled with a carboxyl tail specific MP20 antibody[40] that was diluted in PBS, followed by secondary fluorescein-conjugated antibody (1:200; Santa Cruz Biotechnology, Santa Cruz, CA, USA) for 1 h each[22]. Control sections omitting primary antibody were also prepared. Cell membranes were also labeled with a fluorescein-conjugated wheat germ agglutinin (WGA-TRITC). The WGA-TRITC- was diluted 1:50 in PBS and the sections were incubated for 1 h. After extensive washing in PBS, sections were mounted in anti-fading reagent (Citifluor™, AFI, Canterbury, UK) and viewed with a confocal laser scanning microscope (Leica TCS 4D, Heidelberg, Germany). Hoescht stain was used to highlight the nuclei. Images were pseudo-colored and combined using Adobe Photoshop software.

### Reporting summary

Further information on research design is available in the Nature Portfolio Reporting Summary linked to this article.

### Data availability

The coordinates of the MicroED structure of MP20 were deposited to the PDB and the corresponding potential maps deposited to the EMDB under accession codes 9CVB and EMD-45428, respectively. The raw MicroED data in EER format are available from the corresponding author upon request.

### Code availability

The MicroED tools mrc2smv software is available freely at cryoem.ucla.edu. Additional materials or code are available upon request to the corresponding author.

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

## Acknowledgements

This study was supported by the National Institutes of Health P41GM136508. The Gonen laboratory is supported by funds from the Howard Hughes Medical Institute. P.J.D. is funded by New Zealand Health Research Council Programme Grant.

## Author contributions

A.S. expressed, purified and crystallized human MP20. Y.N.R. performed FPLC analyses. W.J.N. and M.W.M. prepared crystal lamellae, collected and processed the MicroED data and participated in structure analyses. P.J.D. and A.C.G. performed fluorescence analyses on human lenses. T.G. and P.J.D. conceived the project and participated in analyses. All authors participated in writing the manuscript and figure preparation.

## Competing interests

The authors declare no competing interests.
