## [Transparent Peer Review file · Nature Communications]

Structure of the lens MP20 mediated adhesive junction

Corresponding Author: Professor Tamir Gonen

Version 0:

Reviewer comments:

Reviewer #1

(Remarks to the Author)

This is a remarkable study of a small membrane protein within the human ocular lens that has baffled researchers for decades. Although the second most abundant protein in lens fiber cell membranes, its structure and function have been elusive and overshadowed by the prominent aquaporin water channels that are important enough to have received a Nobel award. As a tissue, the lens has the least amount of extracellular space with the close apposition of adjacent fiber cells in the lens interior representing a distinctive feature hypothesized to contribute to the lens optical properties. Close apposition of aquaporin junctions (as well as gap junctions) are important to this unique lens property and early evidence suggested that MP20 may also be involved, although convincing evidence was lacking until this study. The authors have provided high resolution structural data confirming the aggregation of these tetraspanin proteins within one membrane and the close binding of similar clusters across the extracellular space forming a tightly sealed junctional complex. This study is supported by confocal labeling of human tissue to show that, at point of insertion of MP20 into fiber cell membranes, a barrier forms to diffusion of dye within the extracellular space.

General comments.

1. The preparation of the purified protein, the growth of crystals within cubic lipid phases, the labeling of crystals for detection, the isolation into thinned layers and the analysis of the internal structure by MicroED has been expertly performed and described. There are essentially no major concerns and only minor editorial suggestions for this exceedingly well done main portion of the paper.
2. The only major concern is the confocal component described in Figure 4. This portion appears to be an extension of the initial description of the Remodeling Zone (Lim 2009; ref 44) where a similar experimental setup was used and a nearly identical schematic was employed to interpret the results. There are three issues with this component: (1) labels on the diagram are not consistent with the data because the RZ is indicated with an arrow without indicating why and it is displaced compared to the original without explanation; (2) the diagram shows a tangle of membranes (single lines) in the RZ without explanation or how they are related to the hexagonal fiber cells; (3) the dimensions in mm should be μm .
3. The interpretation of the data in Figure 4 is questionable. Most important is that the linear structures seen after MP20 has been incorporated represent adhesive junctions whereas they probably represent much more complex membrane structures. Compare the thickness of the broad face membranes between adjacent developing hexagonal fiber (very fine lines in left panels) to the much broader lines of MP20 labeled structures. It is highly problematic to interpret confocal images in terms of cell structure when cell boundaries are not evident.
4. The most serious issue with Figure 4 is that the RZ has been described about a decade ago using thin-section TEM (Costello et al., EER 116, 2013) where the cellular changes at the beginning of the RZ involve extensive ball-and-socket formations and redistribution of cytoplasmic and membrane components. Indeed, this region looks jumbled in TEM images but not like in the diagram in Figure 4. Furthermore, in the transition zone (TZ) just deeper to the RZ, there appears undulating membranes (not observed in the DF or RZ) and the beginning of cell compaction. Cells in this region are irregular in shape and have thick irregular interfaces (see Fig. 4a in Costello 2013) consistent with the complex pattern of MP20 labeled membranes where only one component is labeled. The predominant patterns in the TZ and deeper in the nucleus are the undulating membranes, which in high magnification TEM images suggest a molecular interpretation where

membranes containing arrays of aquaporins are paired with membranes without protein and, at cross-over points, the association is similar in size and stain distribution predicted for an MP20 adhesive junction (see Fig. 4b, Costello 2013, between second and third loops of the undulations). Although this is a published image, the proposed junction is not labeled so it may be difficult to reference. Such images are not rare when examining human nuclear membranes, which we have been doing for more than a decade, and suggest the possibility that one critical location where thin MP20 controlled adhesive junctions are common is at curvature transitions and my even hold undulating membranes together.

Minor comments.

1. Line 61: omit "of MP20"
2. Line 77: better is "excess material surrounding"
3. Line 355: um should be m
4. Line 483: form should be from
5. Line 488: zone
6. Line 496: should be Figure 2
7. Figure 4: confocal images should have a scale bar

Reviewer #2

(Remarks to the Author)

In the manuscript Structure of the lens MP20 mediated adhesive junction, the authors present the structure of MP20, an intrinsic membrane protein that is the second most abundant membrane protein in the eye lens. MP20 is clearly critical for eye lens development as mutations in this protein cause a multitude of developmental defects. The structure and function of MP20 have remained undetermined for several decades due to the challenges of performing structural biology experiments on small membrane proteins. Indeed, the author's previous work could only resolve this protein to a modest 18 Å by 2D electron crystallography and negative stain EM, revealing that MP20 may exist as a tetramer.

Despite these challenges, the authors were able to form 3D crystals of MP20 in the lipidic cubic phase (LCP), however, they were deemed too small for conventional X-ray crystallography. Utilising correlative focussed ion beam milling and microED the authors were able to obtain diffraction data from these tiny crystals to a resolution of 3.5 Å. MP20 was found to contain 4 transmembrane alpha-helices forming a helical bundle connected by two extracellular loops and one intracellular loop. These extracellular loops seem to play a key role in MP20 function as they allow monomers from opposing bilayers to interact in what the authors term a 'handshake'. This interaction observed in the crystal provides evidence for MP20's role in the formation of adhesive junctions. The length scale of the interaction (11 nm) is also consistent with previously observed lens thin 11 nm junctions.

The authors probe the ability of MP20 to form adhesive interactions both in vitro by vesicle reconstitution assays and in vivo with fixation, sectioning of the human lenses, and labelling with MP20 antibodies to study the localisation and organisation of MP20. The adhesive junctions were also studied by incubating the same lenses with Texas red Dextran dye. Through correlating the images, the authors observed that dye diffusion was restricted by a physical barrier coinciding with the MP20 adhesive junction membrane. The in vivo work provides compelling evidence that MP20 plays a role in maintaining the integrity of the eye lens in line with the observations from the crystal structure.

Overall, this paper presents an excellent example of the use of microED to solve previously intractable structures. Given the abundance of MP20 in the eye lens, knowledge of its structure and function is a very important biological problem. The description of protein expression, FIB milling and targeting, microED and in vivo assays are highly detailed and would be sufficient for replication of these studies. However, there are a few inconsistencies in the manuscript that would need to be addressed before it is suitable for publication.

1. The authors state that MP20 is unsuitable for SPA due to its small size (20 kDa) yet also that the FPLC trace observed indicates it is octameric (160 kDa). This would bring the assembly in line with what is possible with the current state of the art (see recent work from the lab of Sjors Scheres (<https://www.nature.com/articles/s41592-024-02304-8>). Could the authors perhaps expand a bit more on what the challenges would have been for this, perhaps due to the inhomogeneity of the assemblies after purification?

2. Much of the author's conclusions about MP20 lean on the tetrameric assembly observed in the crystals, however, this tetramer is a result of the crystallographic symmetry. Can the authors highlight some of the key interactions between the MP20 monomers that would evidence its existence as a tetramer beyond the symmetry of the crystal? This would be helped if there was a figure outlining the full unit cell and expanded crystal contacts.

3. In the author's previous work, it was stated that both magnesium and native lipids were necessary for the formation of tetrameric assemblies of MP20. Are these ions visible in the structure? How well does the lipid composition of the LCP crystallisation condition reflect the native lens lipids? Were any lipids visible in the final structure?

4. How large are the MP20 crystals that the authors used in this study? The images in Figure 1 are a little unclear, perhaps due to the small size of the crystals. Based on the fluorescent images the crystals appear to be close to 5 - 10 microns in 2 of their dimensions. Whilst small this is not outside of the realms of possibility at modern synchrotron sources: (<https://www.nature.com/articles/s41467-023-39819-1#Sec17> for example). Did the authors try and fail to capture X-ray data

from crystals prepared in the same way as presented in the article? Was the volume of LCP in which they were embedded a large barrier to manipulating the crystals?

5. The authors state “Further, a variety of congenital cataracts are associated with MP20 mutation, but it is not clear what is the structural impact of these mutations have on the protein and how they may affect its function” As a structure is now available it should be possible to map some of these mutations back onto the structure and see what interactions they may interfere with. This could perhaps form a supplementary figure.

6. There is no description of the vesicle reconstitution assay in the methods section. These experiments as presented are very qualitative and somewhat unconvincing. There could be a variety of reasons for vesicle clumping and as only a single image is presented for each condition it is hard to know how representative this is. This section would benefit from further quantification and the inclusion of more representative images in the supplement. Alternatively, this assay could potentially be excluded from the article as it does not add any additional evidence that isn't provided by the in vivo studies.

Minor comments

1. Line 61, 165, 189 – Should in vivo be in italics?

2. In line 114 the completeness should be the measured value of 86.6% rather than an approximation.

3. Would the inclusion of more than 6 datasets have improved the resolution or the completeness of the final dataset?

4. The final r-factors of 33% and 35% are quite high, even for this resolution. What was the reason for the stagnation at such a high value?

5. Line 144 “The interactions appear to be mainly electrostatic packing as several charged and hydroxylated residues can be seen forming adhesive interactions” Figure 3 which is referenced here does not highlight any side chain interactions and is just a cartoon view of the protein, figure s3 however does include this information. Similarly, only one of videos S3 - S6 show residue level information. The correct figure and video should be referenced here.

6. There are missing spaces on line 161 between “TheMP20” and line 175 “membrane.In”

7. There is no explanation for the acronym WGA in line 169

8. The wavelength in Table 1 should be 0.0197 (300 kV), not 0.0176

9. For the microED data collection what was the flux used? Total fluence on the crystal? How large of a region was illuminated? These important details are missing.

10. Alphafold (including alphafold3) has several internal metrics (pIDDT, ipTM and pTM) used to estimate the prediction's accuracy. It would be useful to include these in the text or alongside the supplementary figure showing these models to allow the reader to determine whether they were high-quality predictions that were incorrect or low-quality predictions.

Reviewer #3

(Remarks to the Author)

I co-reviewed this manuscript with one of the reviewers who provided the listed reports. This is part of the Nature Communications initiative to facilitate training in peer review and to provide appropriate recognition for Early Career Researchers who co-review manuscripts

Reviewer #4

(Remarks to the Author)

General points.

In this study by Nicolas and colleagues the authors use MicroED to elucidate the structure of MP20 at 3.5 Å. MP20 is an abundant integral protein of vertebrate lens fiber cells. Mutations in MP20 underlie inherited cataracts in people and animal models emphasizing the importance of this molecule for vision.

As a relatively small intrinsic membrane protein, MP20 has been difficult to study using conventional approaches. The authors are experts in the use of MicroED and deploy that technique to good effect in this study. Using fluorescently-labeled recombinant protein, plasma ion beam milling, and an interesting technique for obtaining continuous diffraction patterns from rotating samples, they show that in lipid-cubic phase, MP20 is organized in tetramers. Handshake interactions between the extracellular loops of two opposed tetramers likely constitute an adhesive junction. The data are convincing and should be of great interest to anyone working on MP20 or other members of the PMP22/EMP/MP20 family, including tight junction proteins such as claudins.

A little less compelling are the dye-labeling experiments in the final part of the paper that seek to link the presence of MP20 in the fiber cell membranes to the formation of the extracellular diffusion barrier between differentiating lens fiber cells. While

it is entirely plausible that MP20 junctions obliterate the extracellular space in the lens interior as the authors suggest, the experiments do not constitute firm evidence for this. Fig 4 at best shows a correlation between membranous MP20 expression and the location of the diffusion barrier. It is also not a very novel experiment – the authors have previously shown the presence of dye-diffusion barriers in rat and human lenses and have previously noted that the appearance of MP20 membrane staining correlates with the location of the barrier. The authors might consider moderating the tone of their conclusions to emphasize that while MP20 junctions are good candidates for the barrier, definitive experimental proof is still wanting.

Specific points.

Abstract.

The abstract overstates a number of the findings. For example, this paper did not show that "MP20 is stored in intracellular vesicles and upon fiber cell maturation was inserted into the membrane restricting extracellular space". The data provided do not allow vesicles to be visualized and no direct evidence is provided that MP20 restricts extracellular space. Functional experiments, perhaps utilizing knockout animals, would be required to test the latter hypothesis.

Line 52. "a distant member of the occludin family" - should that be claudin family?

Please label the x-axis in Figure 1A (time? Fraction number?). Also, why does the purified MP20 run as multimers on SDS-PAGE (when the native protein usually runs as a single ≈ 20 kDa band)?

The authors are fond of the word "dramatic" ("dramatically elongate" "dramatically constricted" "dramatic decrease"), but it's not a very scientific word, and I suggest deleting it. Likewise, please delete "way", as in "way below the limit (line 65).

Line 57. "is found in cytoplasmic vesicles.4,5" the references (4,5) do not refer to cytoplasmic vesicles.

The clumping assay (Fig 3d,e) would be more convincing if an additional control sample containing a non-adherent membrane protein were included.

The best direct evidence for an adhesive role for MP20 perhaps comes from MP20 deficient mouse lenses. Fiber cells from those animals readily dissociate from each other (IOVS 2007, 48, 500-8).

In the methods section it indicates that a range of human lenses were used in this study. However, only one sample is shown in Figure 4. How many lenses were used and did the results vary with age. Also, what was the donor age for the lens shown in the figure?

Line 479. Figure legend. "Human lenses labelled with". The figure only shows one lens.

Figure 4c,d. The extracellular marker (Texas red) seems to be diffusely intracellular in the innermost region of the transition zone. Can the authors comment?

The scale on Fig. 4D. should be in micrometers not millimeters. Also, the diagram appears to have been adapted from an earlier publication. If this is the case the authors should acknowledge the earlier paper.

Reviewer #5

(Remarks to the Author)

This manuscript reports the structure of the MP20 protein, utilizing advanced methods like MicroED and cryo-pFIB milling. This work significantly contributes to our understanding of membrane protein structures and their functions in biological systems, particularly in the lens. Overall, the work design, methodology, data generation, interpretation, discussion, and summary were satisfactory. Moreover, the authors highlighted the need for experimental elucidation rather than prediction. The authors may consider the following points for further refinement of the manuscript.

1) More discussion on potential functional studies of MP20, especially involving mutations, would strengthen the manuscript. These studies could further validate the structural findings and offer deeper insights into MP20's role in the lens.

2) While discussing the implications for other tight junction proteins, consider adding preliminary experimental evidence or suggesting future experiments that could explore these broader impacts.

3) The authors should also emphasize the broader implications of the study. How does this research advance our understanding of the human eye lens or potentially impact clinical practices?

Specific Comments

1. Introduction:

Clarify what is meant by "dramatically elongate" in terms of cellular dimensions or changes.

2. Methods :

Provide more detail on the preparation of samples for MicroED.

3. Results :

Please explain how the findings on MP20 localization can be related to its function in the mature lens.

Version 1:

Reviewer comments:

Reviewer #1

(Remarks to the Author)

See attachment.

Reviewer #2

(Remarks to the Author)

My concerns from the previous version of the manuscript have been addressed satisfactorily and the revised manuscript is much improved. I have three minor points that need to be addressed:

Line 116 The quoted completeness remains unchanged, please quote the measured value of 86.6% rather than an approximate value.

Supplementary Figure 1c is not very clear. It would be helpful to highlight the residues with proper text labels rather than arrows. Which residues are mutated, and to what are they mutated to? They could be highlighted in the sequence displayed in Supplementary Figure 1a.

"When asked for predictive models for a dimer or for the octamer nonsensical assemblies were predicted with a high degree of certainty." This statement is factually inaccurate. When running the MP20 sequence through the Alphafold3 server to obtain tetramer or octamer predictions the confidence in the assemblies (pTM and ipTM) was between 0.2 and 0.4 which is outside of the range of what would be considered a successful prediction (see attached images). To quote the Alphafold3 website "ipTM measures the accuracy of the predicted relative positions of the subunits within the complex. Values higher than 0.8 represent confident high-quality predictions, while values below 0.6 suggest likely a failed prediction". The text needs to be modified to highlight that whilst the assemblies predicted are nonsensical, they are not predicted with high confidence as the authors currently indicate.

See additional attachments.

Reviewer #3

(Remarks to the Author)

Reviewer #4

(Remarks to the Author)

The authors have satisfactorily addressed my concerns.

Reviewer #5

(Remarks to the Author)

The revised manuscript effectively addresses all major reviewer concerns with clear and detailed responses. The structural findings are novel and well-supported by the data. While the in vivo implications of MP20 in restricting extracellular space remain correlative, the authors have appropriately strengthened these claims.

Therefore, I recommend this manuscript for publication, pending minor formatting adjustments

Dear Editor,

We would like to thank you and the five reviewers for considering our manuscript. We are pleased that the reviewers found our work to be of high impact and suitable for publication after revisions. We have carefully considered all the comments and addressed each point in our revised manuscript. We believe that these revisions have strengthened the manuscript, and it is now ready for publication. Our responses appear below in blue following the individual reviewer comments.

REVIEWER COMMENTS

Reviewer #1 (Remarks to the Author):

This is a remarkable study of a small membrane protein within the human ocular lens that has baffled researchers for decades. Although the second most abundant protein in lens fiber cell membranes, its structure and function have been elusive and overshadowed by the prominent aquaporin water channels that are important enough to have received a Nobel award. As a tissue, the lens has the least amount of extracellular space with the close apposition of adjacent fiber cells in the lens interior representing a distinctive feature hypothesized to contribute to the lens optical properties. Close apposition of aquaporin junctions (as well as gap junctions) are important to this unique lens property and early evidence suggested that MP20 may also be involved, although convincing evidence was lacking until this study. The authors have provided high resolution structural data confirming the aggregation of these tetraspanin proteins within one membrane and the close binding of similar clusters across the extracellular space forming a tightly sealed junctional complex. This study is supported by confocal labeling of human tissue to show that, at point of insertion of MP20 into fiber cell membranes, a barrier forms to diffusion of dye within the extracellular space.

General comments.

1. The preparation of the purified protein, the growth of crystals within cubic lipid phases, the labeling of crystals for detection, the isolation into thinned layers and the analysis of the internal structure by MicroED has been expertly performed and described. There are essentially no major concerns and only minor editorial suggestions for this exceedingly well done main portion of the paper.

Thank you.

2. The only major concern is the confocal component described in Figure 4. This portion appears to be an extension of the initial description of the Remodeling Zone (Lim 2009; ref 44) where a similar experimental setup was used and a nearly identical schematic was employed to interpret the results. There are three issues with this component: (1) labels on the diagram are not consistent with the data because the RZ is indicated with an arrow without indicating why and it is displaced compared to the

original without explanation; (2) the diagram shows a tangle of membranes (single lines) in the RZ without explanation or how they are related to the hexagonal fiber cells; (3) the dimensions in mm should be mm.

Yes, the reviewer is correct Figure 4e is a simplified version of the schematic initially published in the study by Lim et al. that in 2009. This confocal based study of the fiber cell morphology in human lenses was the first to document the existence of the remodelling(RZ) and transition (TZ) zones in the lens that are associated with remodelling of fiber cell morphology from the classical smooth hexagonal cell profiles observed in the peripheral differentiating fiber cells to the convoluted membrane morphology seen in mature fiber cells located in the adult nucleus. Our 2009 study promoted Costello et al., in 2013 to investigate this region of the human lens with transmission electron microscopy approaches that preserved the outer cortex of the lens and therefore allowed the remodelling of fiber cell membrane morphology initial observed using confocal microscopy to be resolved at higher resolution. To address the reviewers concerns with our simplification of Figure 4E in which we show the RZ and TZ's as more of a continuum than discrete zones we have redrawn Figure 4E. In this redrawn Figure 4E we have also incorporated the findings from Costello's TEM study which provides the resolution to show what is happening to the fiber cell membrane morphology at the ultrastructure level before and after the membrane insertion of MP20. This revised figure also helps to address the comments 3 and 4 from this referee which are addressed below. In this revised Figure we have address the error evident in the scale bar.

3. The interpretation of the data in Figure 4 is questionable. Most important is that the linear structures seen after MP20 has been incorporated represent adhesive junctions whereas they probably represent much more complex membrane structures. Compare the thickness of the broad face membranes between adjacent developing hexagonal fiber (very fine lines in left panels) to the much broader lines of MP20 labeled structures. It is highly problematic to interpret confocal images in terms of cell structure when cell boundaries are not evident.

Agreed. Hence to address this point and to provide higher resolution data on the ultrastructure of fiber cell membranes than can be provided by confocal microscopy we have drawn on the TEM data provided by Costello et al., 2013 that mapped the changes in membrane morphology throughout this same region of the human lens. These changes in membrane ultrastructure have been integrated into the new Figure 4E.

4. The most serious issue with Figure 4 is that the RZ has been described about a decade ago using thin-section TEM (Costello et al., EER 116, 2013) where the cellular changes at the beginning of the RZ involve extensive ball-and-socket formations and redistribution of cytoplasmic and membrane components. Indeed, this region looks jumbled in TEM images but not like in the diagram in Figure 4. Furthermore, in the transition zone (TZ) just deeper to the RZ, there appears undulating membranes (not observed in the DF or RZ) and the beginning of cell compaction. Cells in this region are irregular in shape and have thick irregular interfaces (see Fig. 4a in Costello 2013) consistent with the complex pattern of MP20 labeled membranes where only one component is labeled. The predominant patterns in the TZ and deeper in the nucleus are the undulating membranes, which in high magnification TEM images suggest a molecular interpretation where membranes containing arrays of aquaporins are paired with membranes without protein and, at cross-over points, the association is similar in size and stain distribution predicted for an MP20 adhesive junction (see Fig. 4b, Costello 2013, between second and third loops of the undulations). Although this is a published image, the proposed junction is not labeled so it may be difficult to reference. Such images are not rare when examining human nuclear membranes, which we have been doing for more than a decade, and suggest the possibility that one critical location where thin MP20 controlled adhesive junctions are common is at curvature transitions and my even hold undulating membranes together.

As stated above Costello's study was prompted by our initial confocal study that for the first time showed the dramatic remodelling of differentiating fiber cells in the outer cortex of the lens. Costello's TEM study basically confirmed the existence of the RZ and TZ zones while providing the necessary resolution to visualise the changes in membrane ultrastructure that could be not resolved by our confocal based imaging approach. Costello's images show that the distortion of the membrane morphology seen in confocal images through the remodelling zone is caused by the appearance of numerous membrane undulations that have previously been characterised as wavy junctions (Costello MJ, McIntosh TJ, Robertson JD. Distribution of gap junctions and square

array junctions in the mammalian lens. *Investigative ophthalmology & visual science* 1989;30:975-989. PMID: 2722452.). *Costello does indeed speculate that these junctions maybe formed by AQP0 but the 11 nm size of the MP20 junctions obtained in the current study and the observation that these membrane undulations only form after MP20 moves from a cytoplasmic location to the plasma membrane suggests MP20 may also be involved in their formation along with AQP0. It should also be noted that no one protein is likely to drive the extensive remodelling seen in this area and Costello comments on the potential role played by the cytoskeleton and changes to the distribution of crystallins during the remodelling process that transforms differentiating fiber cells into mature fibers in the adult nucleus of the lens. Hence further studies such as immuno-EM or super resolution microscopy will be required to determine whether MP20 contributes to these membrane undulations following its insertion into the membrane and whether these membrane undulations are the structures that restricts the penetration of the extracellular marker Texas red further into the lens.*

Minor comments.

1. Line 61: omit “of MP20”

Done

2. Line 77: better is “excess material surrounding “

Agreed. This has been changed.

3. Line 355: um should be mm

This has been corrected in the new figure.

4. Line 483: form should be from

Done

5. Line 488: zone

Done

6. Line 496: should be Figure 2

Done

7. Figure 4: confocal images should have a scale bar

Done

Reviewer #2 (Remarks to the Author):

In the manuscript Structure of the lens MP20 mediated adhesive junction, the authors present the structure of MP20, an intrinsic membrane protein that is the second most abundant membrane protein in the eye lens. MP20 is clearly critical for eye lens development as mutations in this protein cause a multitude of developmental defects, The structure and function of MP20 have remained undetermined for several decades

due to the challenges of performing structural biology experiments on small membrane proteins. Indeed, the author's previous work could only resolve this protein to a modest 18 Å by 2D electron crystallography and negative stain EM, revealing that MP20 may exist as a tetramer.

Despite these challenges, the authors were able to form 3D crystals of MP20 in the lipidic cubic phase (LCP), however, they were deemed too small for conventional X-ray crystallography. Utilising correlative focussed ion beam milling and microED the authors were able to obtain diffraction data from these tiny crystals to a resolution of 3.5 Å. MP20 was found to contain 4 transmembrane alpha-helices forming a helical bundle connected by two extracellular loops and one intracellular loop. These extracellular loops seem to play a key role in MP20 function as they allow monomers from opposing bilayers to interact in what the authors term a 'handshake'. This interaction observed in the crystal provides evidence for MP20's role in the formation of adhesive junctions. The length scale of the interaction (11 nm) is also consistent with previously observed lens thin 11 nm junctions.

The authors probe the ability of MP20 to form adhesive interactions both in vitro by vesicle reconstitution assays and in vivo with fixation, sectioning of the human lenses, and labelling with MP20 antibodies to study the localisation and organisation of MP20. The adhesive junctions were also studied by incubating the same lenses with Texas red Dextran dye. Through correlating the images, the authors observed that dye diffusion was restricted by a physical barrier coinciding with the MP20 adhesive junction membrane. The in vivo work provides compelling evidence that MP20 plays a role in maintaining the integrity of the eye lens in line with the observations from the crystal structure.

Overall, this paper presents an excellent example of the use of microED to solve previously intractable structures. Given the abundance of MP20 in the eye lens, knowledge of its structure and function is a very important biological problem. The description of protein expression, FIB milling and targeting, microED and in vivo assays are highly detailed and would be sufficient for replication of these studies. However, there are a few inconsistencies in the manuscript that would need to be addressed before it is suitable for publication.

1. The authors state that MP20 is unsuitable for SPA due to its small size (20 kDa) yet also that the FPLC trace observed indicates it is octameric (160 kDa). This would bring the assembly in line with what is possible with the current state of the art (see recent work from the lab of Sjors Scheres (<https://www.nature.com/articles/s41592-024-02304-8>)). Could the authors perhaps expand a bit more on what the challenges would have been for this, perhaps due to the inhomogeneity of the assemblies after purification?

The lipid membrane, from the lipidic cubic phase, enables the MP20 tetramers and later junction to form. In the context of the detergent the assembly breaks down to

monomers. This type of behavior is well known for multimeric membrane proteins. In this case, SPA was attempted by Gonen 20 years ago and only tiny monomers, possibly dimers, were observed. However, in the context of a membrane the larger assembly was seen (DOI 10.1016/j.jmb.2007.09.001). Hence, this is a beautiful example where SPA was not suitable despite the larger size of the assembly.

2. Much of the author's conclusions about MP20 lean on the tetrameric assembly observed in the crystals, however, this tetramer is a result of the crystallographic symmetry. Can the authors highlight some of the key interactions between the MP20 monomers that would evidence its existence as a tetramer beyond the symmetry of the crystal? This would be helped if there was a figure outlining the full unit cell and expanded crystal contacts.

The oligomeric assembly of MP20 is well described and explained in the context of the lipid membrane. See reference above. Given the higher resolution we have added a description of the monomer-monomer interface.

3. In the author's previous work, it was stated that both magnesium and native lipids were necessary for the formation of tetrameric assemblies of MP20. Are these ions visible in the structure? How well does the lipid composition of the LCP crystallisation condition reflect the native lens lipids? Were any lipids visible in the final structure?

Lipids and magnesium are present in our preparations here also, however, we do not have sufficiently high enough resolution to identify 1. Which lipids are bound to MP20 and how; and 2. What role magnesium may play. Significantly higher resolution would be required and will be the subject of future work. In comparison no lipids were observed in the 3A AQP0 structure but a complete lipid bilayer seen at 1.9A (See 10.1038/nature02503. and doi: 10.1038/nature04321).

4. How large are the MP20 crystals that the authors used in this study? The images in Figure 1 are a little unclear, perhaps due to the small size of the crystals. Based on the fluorescent images the crystals appear to be close to 5 - 10 microns in 2 of their dimensions. Whilst small this is not outside of the realms of possibility at modern synchrotron sources: (<https://www.nature.com/articles/s41467-023-39819-1#Sec17> for example). Did the authors try and fail to capture X-ray data from crystals prepared in the same way as presented in the article? Was the volume of LCP in which they were embedded a large barrier to manipulating the crystals?

The MP20 crystals in the LCP were typically ~2x2um in x-y but incredibly thin at <1um in z. Although we attempted X-ray diffraction these attempts were unsuccessful. This is similar to past experiments where despite major effort X-ray diffraction was not obtained. The volume of the drop is ~2ul and crystals are only clearly visible in fluorescence making handling of the crystals difficult. XFELs were not attempted because the crystals were too sparse and we could not grow large batches.

5. The authors state “Further, a variety of congenital cataracts are associated with MP20 mutation, but it is not clear what is the structural impact of these mutations have on the protein and how they may affect its function” As a structure is now available it should be possible to map some of these mutations back onto the structure and see what interactions they may interfere with. This could perhaps form a supplementary figure.

We have added a supplementary figure showing where these mutations map to our structure.

6. There is no description of the vesicle reconstitution assay in the methods section. These experiments as presented are very qualitative and somewhat unconvincing. There could be a variety of reasons for vesicle clumping and as only a single image is presented for each condition it is hard to know how representative this is. This section would benefit from further quantification and the inclusion of more representative images in the supplement. Alternatively, this assay could potentially be excluded from the article as it does not add any additional evidence that isn't provided by the in vivo studies.

We agree. This figure and study has been removed from the manuscript.

Minor comments

1. Line 61, 165, 189 – Should in vivo be in italics?

We just leave it. Nature doesn't italicize latin but most other journals do – we let the editors decide.

2. In line 114 the completeness should be the measured value of 86.6% rather than an approximation.

Fixed.

3. Would the inclusion of more than 6 datasets have improved the resolution or the completeness of the final dataset?

The crystals are highly susceptible to radiation damage so obtaining higher resolution has not been possible yet. Moreover, the thinness of the crystals made merging and refinement quite challenging due to non-isomorphism in the unit cells. We chose this number after a brute force attempt at all possible merges giving the best statistics. Merging additional datasets did not improve the completeness, and just resulted in worse maps and worse statistics, so this number of datasets gave us the best merging statistics without sacrificing completeness. We note that the crystals being sub-micron in one dimension makes finding the missing orientation quite challenging.

4. The final r-factors of 33% and 35% are quite high, even for this resolution. What was the reason for the stagnation at such a high value?

We agree. We suspect that lipids are part of the unit cell but we cant see them due to the poor resolution. Hence the refinement cannot proceed beyond what we have presented unless we could obtain higher resolution and higher completeness. We have not been able to do either so far largely because of the challenges given in #3.

5. Line 144 “The interactions appear to be mainly electrostatic packing as several charged and hydroxylated residues can be seen forming adhesive interactions” Figure 3 which is referenced here does not highlight any side chain interactions and is just a cartoon view of the protein, figure s3 however does include this information. Similarly, only one of videos S3 - S6 show residue level information. The correct figure and video should be referenced here.

We have changed this to correctly reference S3 and videos S3-S6.

6. There are missing spaces on line 161 between “TheMP20” and line 175 “membrane.In”

Fixed

7. There is no explanation for the acronym WGA in line 169 - *it is Wheatgerm Agglutinin*

We have spelled this out. Thank you.

8. The wavelength in Table 1 should be 0.0197 (300 kV), not 0.0176

Fixed.

9. For the microED data collection what was the flux used? Total fluence on the crystal? How large of a region was illuminated? These important details are missing.

We agree. This information now appears in the methods section of the paper.

10. Alphafold (including alphafold3) has several internal metrics (pIDDT, ipTM and pTM) used to estimate the prediction’s accuracy. It would be useful to include these in the text or alongside the supplementary figure showing these models to allow the reader to determine whether they were high-quality predictions that were incorrect or low-quality predictions.

This might be useful, but will highlight a different issue. pIDDT etc are reported at the monomer level. The monomer prediction is always very high per residue. Our highlights show that the prediction of the multimer is incorrect. We have edited the manuscript to highlight this fact.

Reviewer #3 (Remarks to the Author):

I co-reviewed this manuscript with one of the reviewers who provided the listed reports. This is part of the Nature Communications initiative to facilitate training in peer review and to provide appropriate recognition for Early Career Researchers who co-review manuscripts

Thank you for contributing to this work by reviewing our manuscript.

Reviewer #4 (Remarks to the Author):

General points.

In this study by Nicolas and colleagues the authors use MicroED to elucidate the structure of MP20 at 3.5 Å. MP20 is an abundant integral protein of vertebrate lens fiber cells. Mutations in MP20 underlie inherited cataracts in people and animal models emphasizing the importance of this molecule for vision.

As a relatively small intrinsic membrane protein, MP20 has been difficult to study using conventional approaches. The authors are experts in the use of MicroED and deploy that technique to good effect in this study. Using fluorescently-labeled recombinant protein, plasma ion beam milling, and an interesting technique for obtaining continuous diffraction patterns from rotating samples, they show that in lipid-cubic phase, MP20 is organized in tetramers. Handshake interactions between the extracellular loops of two opposed tetramers likely constitute an adhesive junction. The data are convincing and should be of great interest to anyone working on MP20 or other members of the PMP22/EMP/MP20 family, including tight junction proteins such as claudins.

A little less compelling are the dye-labeling experiments in the final part of the paper that seek to link the presence of MP20 in the fiber cell membranes to the formation of the extracellular diffusion barrier between differentiating lens fiber cells. While it is entirely plausible that MP20 junctions obliterate the extracellular space in the lens interior as the authors suggest, the experiments do not constitute firm evidence for this. Fig 4 at best shows a correlation between membranous MP20 expression and the location of the diffusion barrier. It is also not a very novel experiment – the authors have previously shown the presence of dye-diffusion barriers in rat and human lenses and have previously noted that the appearance of MP20 membrane staining correlates with the location of the barrier. The authors might consider moderating the tone of their conclusions to emphasize that while MP20 junctions are good candidates for the barrier, definitive experimental proof is still wanting.

We thank the reviewer for their comments. Although we have published similar work on rodent lenses and some bits on human lenses, adding the immunofluorescence study here provides important context. We have moderated the tone of our conclusions as requested. We also added the report from rodents that show fiber cells from MP20 deficient mouse lenses readily dissociate from each other (IOVS 2007, 48, 500-8).

Specific points.

Abstract.

The abstract overstates a number of the findings. For example, this paper did not show that "MP20 is stored in intracellular vesicles and upon fiber cell maturation was

inserted into the membrane restricting extracellular space". The data provided do not allow vesicles to be visualized and no direct evidence is provided that MP20 restricts extracellular space. Functional experiments, perhaps utilizing knockout animals, would be required to test the latter hypothesis.

We have rephrased this sentence to more accurately reflect our current observations

"MP20 is initially localized to the cytoplasm in differentiating fiber cells but upon fiber cell maturation is inserted into the plasma membrane, an event that coincides with a remodeling of the membrane morphology and a restriction of the diffusion of extracellular tracers into the lens".

Line 52. "a distant member of the occludin family" - should that be claudin family?

Sorry for the mistake. MP20 is a distant member of the Claudin family. We have corrected this mistake in the revised version.

Please label the x-axis in Figure 1A (time? Fraction number?). Also, why does the purified MP20 run as multimers on SDS-PAGE (when the native protein usually runs as a single ≈ 20 kDa band)?

X axis is ml. This was added to the figure.

Normally proteins are boiled in SDS prior to running an SDS PAGE. However, membrane proteins could aggregate so we never boil the sample before running the gel. As a result, some multimers can be seen on the SDS Page especially in the western blot.

The authors are fond of the word "dramatic" ("dramatically elongate" "dramatically constricted" "dramatic decrease"), but its not a very scientific word, and I suggest deleting it. Likewise, please delate "way", as in "way below the limit (line 65).

Done

The clumping assay (Fig 3d,e) would be more convincing if an additional control sample containing a non-adherent membrane protein were included.

We have removed this figure from the paper as suggested by another reviewer.

The best direct evidence for an adhesive role for MP20 perhaps comes from MP20 deficient mouse lenses. Fiber cells from those animals readily dissociate from each other (IOVS 2007, 48, 500-8).

In the methods section it indicates that a range of human lenses were used in this study. However, only one sample is shown in Figure 4. How many lenses were used and did the results vary with age. Also, what was the donor age for the lens shown in the figure?

3 human lens was used for the fluorescent study of the insertion of MP20 in the membrane of lens cells (Figure 4). Biological replicates = 3 human lenses (63y M, 46y M, 71 y M) were used for the labelling.

Line 479. Figure legend. "Human lenses labelled with". The figure only shows one lens.

Corrected

Figure 4c,d. The extracellular marker (Texas red) seems to be diffusely intracellular in the innermost region of the transition zone. Can the authors comment?

We have performed the Texas red-dextran incubation experiments on different species of lenses using a range of incubation times to confirm that the penetration of the extracellular tracer into the lens is not diffusion limited. At some of the longer incubation times used we have on occasion seen that Texas red-dextran is taken up by the fiber cells. We concluded that this uptake was due to the transcytosis of Texas red-dextran as this process has been observed in lenses that were incubate for longer periods in organ culture (Boyle DL, Carman P, Takemoto L. Translocation of macromolecules into whole rat lenses in culture. Mol Vis. 2002 Jul 10;8:226-34)

The scale on Fig. 4D. should be in micrometers not millimeters. Also, the diagram appears to have been adapted from an earlier publication. If this is the case the authors should acknowledge the earlier paper.

We have redrawn Figure 4 which addresses these comments

Reviewer #5 (Remarks to the Author):

This manuscript reports the structure of the MP20 protein, utilizing advanced methods like MicroED and cryo-pFIB milling. This work significantly contributes to our understanding of membrane protein structures and their functions in biological systems, particularly in the lens. Overall, the work design, methodology, data generation, interpretation, discussion, and summary were satisfactory. Moreover, the authors highlighted the need for experimental elucidation rather than prediction.

The authors may consider the following points for further refinement of the manuscript.

1) More discussion on potential functional studies of MP20, especially involving mutations, would strengthen the manuscript. These studies could further validate the structural findings and offer deeper insights into MP20's role in the lens.

We agree with the reviewer that our paper is just the initial report of MP20 function, clearly identifying the protein as a major tight junction forming protein in the lens. But much more needs to be done such as channel recordings, mutagenesis and better crystallization to increase the resolution. Such studies are well beyond the scope of the current report but are things we plan to pursue.

2) While discussing the implications for other tight junction proteins, consider adding preliminary experimental evidence or suggesting future experiments that could explore these broader impacts.

Additional comments were added to the concluding remarks.

3) The authors should also emphasize the broader implications of the study. How does this research advance our understanding of the human eye lens or potentially impact clinical practices?

We have significantly expanded our conclusions to point to these greater impacts.

Specific Comments

1. Introduction: Clarify what is meant by "dramatically elongate" in terms of cellular dimensions or changes.

We deleted the word dramatically.

2. Methods :Provide more detail on the preparation of samples for MicroED.

We have updated our MicroED methods sections as suggested by another reviewer.

3. Results :Please explain how the findings on MP20 localization can be related to its function in the mature lens.

We have added a section in the discussion as requested. Here we suggest that MP20 supports lens architecture and plasticity by acting as an adhesive protein.

We would like to thank the reviewers and editor for their helpful comments. We have addressed all the remaining few reviewer comments and believe the paper is now ready for publication. Below each point is addressed point by point with our responses in **blue**.

Reviewer #1 (Remarks to the Author):

The paragraph discussing the importance of the MP20 junctions in the Concluding Remarks, lines 237-249, is a valuable addition, especially the third sentence "The close apposition of fiber cells, facilitated by these junctions, minimizes light scattering and maintains the lens's optical properties." This discussion would benefit from a literature reference to support the assertions. I suggest that the paper already mentioned above, Costello et al., EER 87, 147-158 (2008), is appropriate here, because it includes a theoretical treatment of scattering from multiple thin interfaces. By employing the Herpin formalism for calculating the scattering from thousands of layers, analogous to the undulating membranes in a human lens nucleus, it was shown that transparency could be maintained if the layers were all sufficiently thin, about 30 nm for normal lenses. Although the calculations were done for light hitting perpendicular to the membranes, it can be generalized for light striking membranes at multiple angles simulating the clusters of highly undulating membranes in the nuclear core. The importance of relating these studies to cataract formation is also discussed in this paper by showing that the predicted light scattering increases dramatically if the extracellular space increases, as in some types of cataracts.

RESPONSE: We have added the reference to Costello et al. EER 87 (2008) to this claim and thank the reviewer for the suggestion.

Reviewer #2 (Remarks to the Author):

My concerns from the previous version of the manuscript have been addressed satisfactorily and the revised manuscript is much improved. I have three minor points that need to be addressed:

RESPONSE: We thank the reviewer for their feedback.

Line 116 The quoted completeness remains unchanged, please quote the measured value of 86.6% rather than an approximate value.

DONE

Supplementary Figure 1c is not very clear. It would be helpful to highlight the residues with proper text labels rather than arrows. Which residues are mutated, and to what are they mutated to? They could be highlighted in the sequence displayed in Supplementary Figure 1a.

Our final version of the Figure, Supplementary Figure, and Supplementary Movie should be of higher quality than the compressed versions in the original submission and be fully visible.

"When asked for predictive models for a dimer or for the octamer nonsensical assemblies were predicted with a high degree of certainty." This statement is factually inaccurate. When running the MP20 sequence through the AlphaFold3 server to obtain tetramer or octamer predictions the confidence in the assemblies (pTM and ipTM) was between 0.2 and 0.4 which is outside of the range

of what would be considered a successful prediction (see attached images). To quote the Alphafold3 website "ipTM measures the accuracy of the predicted relative positions of the subunits within the complex. Values higher than 0.8 represent confident high-quality predictions, while values below 0.6 suggest likely a failed prediction". The text needs to be modified to highlight that whilst the assemblies predicted are nonsensical, they are not predicted with high confidence as the authors currently indicate.

RESPONSE: We modified the statement to: "When asked for predictive models for a dimer or for the octamer nonsensical assemblies were predicted, though with a low degree of certainty." Which we agree is more correct.

Reviewer #3 (Remarks to the Author):

RESPONSE: We thank the reviewer for their feedback.

Reviewer #4 (Remarks to the Author):

The authors have satisfactorily addressed my concerns.

RESPONSE: We thank the reviewer for their feedback.

Reviewer #5 (Remarks to the Author):

The revised manuscript effectively addresses all major reviewer concerns with clear and detailed responses. The structural findings are novel and well-supported by the data. While the in vivo implications of MP20 in restricting extracellular space remain correlative, the authors have appropriately strengthened these claims.

Therefore, I recommend this manuscript for publication, pending minor formatting adjustments.

RESPONSE: We thank the reviewer for their feedback.

Reviewer #1 attachment:

1. My initial review was mainly concerned with the confocal experiments described in Figure 4. The revision contains a modified diagram in Figure 4E that accurately displays the ultrastructural changes in the TEM study by Costello et al., EER 116, 411-418 (2013). This new figure is an important improvement that shows the complexity and prominence of the undulating fiber cell membranes seen in the TZ and throughout the lens nucleus. Considerable evidence from a variety of labs shows extensive molecular details of the role of AQP0 in undulating membranes, which is more than speculation (rebuttal to comment 4, line 11), culminating in the thin section TEM study Costello et al., EER 87, 147-158 (2008). Linear sheets of AQP0 membranes associate to form pentalamellar junctions 14nm thick, called thin symmetric junctions, distinguished from thicker 16nm gap junctions. (These junctions can also be distinguished with freeze-fracture EM, because gap junctions have dense irregular arrays of intramembranous particles, whereas thin symmetric junctions show 6.6nm square arrays of AQP0 proteins.) As these membranes mature in the deep cortex, they take on low-amplitude undulations that increase in amplitude as they age, as well as redistribute the AQP0 arrays to pair with protein-poor membranes (devoid of intramembrane particles) forming thin asymmetric junctions about 11-12nm thick. The transitions in curvature in high-amplitude undulations often show adhesive contacts as 14nm thin symmetric junctions. It is at these curvature transitions that sometimes a thinner 11nm symmetric junction can be visualized. Two examples are present in published images from the 2008 (Fig 8B, between the ECS regions on the right) and 2013 (Fig 4B, between first and second undulations) papers. At high magnification, it is possible to identify thin paired membranes that are distinct from the other thicker junctions in the same images and these two examples may represent MP20 adhesive junctions from native human membranes as preliminary evidence (red boxes, white arrows, 11nm thickness). Further labeling studies will be needed to confirm this identification and localization of MP20 within the undulating membranes throughout the nuclear core. This observation also provides a role for MP20 that is consistent with its high density as the second most abundant lens membrane protein, because it could be present at most transitions in curvature, possibly stabilizing the membranes.

2. The paragraph discussing the importance of the MP20 junctions in the Concluding Remarks, lines 237-249, is a valuable addition, especially the third sentence "The close apposition of fiber cells, facilitated by these junctions, is minimizes light scattering and maintains the lens's optical properties." This discussion would benefit from a literature reference to support the assertions. I suggest that the paper already mentioned above, Costello et al., EER 87, 147-158 (2008), is appropriate here, because it includes a theoretical treatment of scattering from multiple thin interfaces. By employing the Herpin formalism for calculating the scattering from thousands of layers, analogous to the undulating membranes in a human lens nucleus, it was shown that transparency could be maintained if the layers were all sufficiently thin, about 30 nm for normal lenses. Although the calculations were done for light hitting perpendicular to the membranes, it can be generalized for light striking membranes at multiple angles simulating the clusters of highly undulating membranes in the nuclear core. The importance of relating these studies to cataract formation is also discussed in this paper by showing that the predicted light scattering increases dramatically if the extracellular space increases, as in some types of cataracts.

B

Fig 8B, EER 87, 147 (2008)

Fig 4B, EER 116, 411

Reviewer #2 attachments:

Very high (pIDDT > 90)

Confident (90 > pIDDT > 70)

Low (70 > pIDDT > 50)

Very low (pIDDT < 50)

ipTM = 0.2 pTM = 0.27 learn more

Very high (pIDDT > 90)

Confident (90 > pIDDT > 70)

Low (70 > pIDDT > 50)

Very low (pIDDT < 50)

ipTM = 0.21

pTM = 0.4

learn more